# Stochastic Theory of Discrete Binary Fragmentation—Kinetics and Thermodynamics

**DOI:** 10.3390/e24020229

**Published:** 2022-01-31

**Authors:** Themis Matsoukas

**Affiliations:** Department of Chemical Engineering, Pennsylvania State University, State College, PA 16802, USA; txm11@psu.edu; Tel.: +1-814-863-2002

**Keywords:** fragmentation, shattering, phase transitions, statistical thermodynamics, partition function

## Abstract

We formulate binary fragmentation as a discrete stochastic process in which an integer mass *k* splits into two integer fragments *j*, k−j, with rate proportional to the fragmentation kernel $F_{j,k-j}$. We construct the ensemble of all distributions that can form in fixed number of steps from initial mass *M* and obtain their probabilities in terms of the fragmentation kernel. We obtain its partition function, the mean distribution and its evolution in time, and determine its stability using standard thermodynamic tools. We show that shattering is a phase transition that takes place when the stability conditions of the partition function are violated. We further discuss the close analogy between shattering and gelation, and between fragmentation and aggregation in general.

## 1. Introduction

Binary fragmentation is the splitting of a cluster into two. It is the inverse of binary aggregation, in which two clusters merge into a new one. Aggregation and fragmentation are two of the most common mechanisms in population balances. Both processes are irreversible, but in opposite directions: to ever increasing sizes in aggregation, and ever decreasing sizes in breakup. In addition, both processes are capable of exhibiting a remarkable dynamic behavior manifested in the breakdown of mean-field models that describe them. In aggregation, this behavior is known as *gelation*, and is characterized by the emergence of a giant cluster with infinite mass (the “gel”) that accounts for a finite fraction of the total mass [1,2,3,4]. An analogous behavior is known to occur in fragmentation in the form of a ghost phase of clusters with zero mass (“dust”) that contain a finite fraction of the total mass [5,6]. This process of “shattering” has evoked analogies to phase transitions and motivated numerous investigations of the kinetic equations of fragmentation and the conditions that may result to this unique behavior [7,8,9,10,11,12,13,14,15].

The usual approach to the study of fragmentation is through the fragmentation equation, an integro-differential equation for the mean cluster distribution [16]. In reality, fragmentation is a discrete stochastic processes: a discrete cluster disintegrates into discrete fragments. The distribution of clusters itself is a stochastic variable, whose evolution in time represents one of many possible trajectories in the space of feasible distributions. The classical treatment presumes that the solution to the fragmentation equation is the same distribution that would be obtained by averaging over all trajectories. Whether this assumption is correct or not requires a formal treatment of the stochastic problem, but one case where this is clearly not true is shattering. At its most elementary level, fragmentation is mass conserving; the loss of mass conservation in the kinetic equation is indication that this equation no longer tracks the true mean distribution. In this paper, we formulate fragmentation as a discrete stochastic process. The distribution of clusters is itself a random variable and transitions in time. The set of all possible distributions that can be reached in a fixed number of fragmentation events forms the fragmentation ensemble. The goal of the theory is to assign probabilities to the distributions of the ensemble based on the breakup law. Once these probabilities are assigned, the mean distribution may be obtained by direct averaging.

This approach is most general and encompasses the classical kinetic treatment as a special result. It has the further advantage that it makes contact with statistical mechanics. The set of feasible of distributions is a thermodynamic ensemble: its most probable distribution in the asymptotic limit is overwhelmingly more probable than all others and maximizes a functional similar to entropy. This connection offers the tools to treat shattering as a formal phase transition using the stability criteria of phase equilibrium. We have previously applied this approach to binary aggregation and shown that gelation is a formal phase transition in the thermodynamic sense [4,17,18,19]. Here, we formulate the statistical mechanics of the fragmentation ensemble and show that shattering is a phase transition mathematically symmetric to gelation.

The paper is organized as follows. In Section 2, we summarize the elements of the ensemble theory, which forms the basis of the stochastic treatment. In Section 3, we construct the ensembles of discrete binary fragmentation and obtain the kinetic equations that relate transitions between generations. In Section 4, we apply the theory to random fragmentation and obtain the complete solution in closed form. We discuss shattering in Section 5, using stability analysis, and finally, we summarize the results in Section 6.

## 2. Statistical Thermodynamics of the Cluster Ensemble

We begin with a brief summary of the thermodynamic theory that forms the basis of the study. A detailed exposition is given elsewhere [17,20]. We consider a discrete distribution of clusters $n=(n_{1},n_{2}\cdots )$, where $n_{i}$ is the number of clusters with mass i=1,2⋯. The number of cluster in n is *N* and their total mass is *M*:(1)$$\begin{array}{c} \sum_{i}n_{i}=N,\;\sum_{i}in_{i}=M \end{array}$$ We construct the ensemble $E_{M,N}$ of every possible distribution that satisfies these conditions; this ensemble contains all possible ways to partition integer mass *M* into *N* clusters. We assign probabilities to the distributions of the ensemble according to
(2)$$\begin{array}{c} P\left(n\right)=\frac{n!W(n)}{\Omega_{M,N}}, \end{array}$$
where n! is the multinomial coefficient,
(3)$$\begin{array}{c} n=\frac{N!}{n_{1}!\;n_{2}!\cdots }; \end{array}$$
W(n) is a functional of distribution n that embodies the details of the stochastic process and which will remain unspecified until we consider specific models of fragmentation; and $\Omega_{M,N}$ is the partition function, a factor that ensures proper normalization of probabilities,
(4)$$\begin{array}{c} \Omega_{M,N}=\sum_{n\in E_{M,N}}P\left(n\right). \end{array}$$ We refer to P(n) as microcanonical probability and to $\Omega_{M,N}$ as its microcanonical partition function.

The most probable distribution $n^{*}$ in the ensemble is obtained by maximizing logP(n) with respect to n under the two constraints in Equation (Equation 1). The result of this maximization is
(5)$$\begin{array}{c} \frac{n_{k}^{*}}{N}=w_{k}^{*}\frac{e^{-\beta k}}{q}, \end{array}$$
with β (inverse “temperature”), *q* (“canonical” partition function) and $w_{k}^{*}$ (“multiplicity” of cluster size *k*) given by
(6)$$\begin{array}{c} \beta ={\left(\frac{\partial \log\Omega_{M,N}}{\partial M}\right)}_{N},\;\log q={\left(\frac{\partial \log\Omega_{M,N}}{\partial N}\right)}_{M},\;\log w_{k}^{*}={\left(\frac{\partial \log W(n^{*})}{\partial n_{k}^{*}}\right)}_{n_{i\neq k}^{*}}. \end{array}$$ In the asymptotic limit M>N≫1 the most probable distribution becomes overwhelmingly more probable. All properties of the ensemble in the asymptotic limit reduce to the corresponding property of the most probable distribution. The fact that $n^{*}$ maximizes the microcanonical probability leads to the following inequality
(7)$$\begin{array}{c} \Omega_{M,N}\geq H\left(n\right)+\log W\left(n\right), \end{array}$$
where H(n) is the Shannon entropy of distribution
(8)$$\begin{array}{c} H\left(n\right)=-\sum_{i}n_{i}\log\frac{n_{i}}{N}. \end{array}$$ The inequality in Equation (Equation 7) is a generalized statement of the second law: it applies to all distributions in the ensemble and becomes an exact equality only for $n=n^{*}$.

Equations (Equation 2)–(Equation 7) provide a treatment in the language of statistical mechanics of the ensemble of distributions defined by the two constraints in Equation (Equation 1). Next, we apply this formulation to stochastic fragmentation.

## 3. Binary Fragmentation

### 3.1. Fragmentation Kernel

In binary fragmentation, mass *k* breaks into a pair of fragments with masses k−j and *j*. All masses are larger than or equal to 1 (monomers do not break any further). The process can be represented in the form of a reaction
(9)$$\begin{array}{c} \left(k\right)\overset{F_{k-j,j}}{\to }\left(k-j\right)+\left(j\right), \end{array}$$
where $F_{k-j,j}$ is the rate of formation of ordered pair of fragments (k−j,j) and satisfies the symmetry condition $F_{j,k-j}=F_{k-j,j}$. A number of related functions may be defined based on this kernel. The breakage rate of mass *k* is
(10)$$\begin{array}{c} a_{k}=\sum_{j=1}^{k-1}F_{k-j,j}, \end{array}$$
and the number of fragments with size *j* produced from particle mass *k* is
(11)$$\begin{array}{c} b\left(j|k\right)=\frac{2F_{k-j,j}}{\sum_{j=1}^{k-1}F_{k-j,j}}. \end{array}$$ The mean fragmentation kernel within distribution n is the mean value of $F_{i,j}$ among all breakup events in the distribution:(12)$$\begin{array}{c} F\left(n\right)=\frac{1}{M-N}\sum_{i=2}^{\infty }\sum_{j=1}^{i-1}n_{i}F_{i-j,j}, \end{array}$$
where M−N is the total number of fragmentation events in distribution n. The ensemble average fragmentation kernel in ensemble $E_{M,N}$ is the mean value of the fragmentation kernel $F_{i,j}$ over all possible fragmentation events in the ensemble:(13)$$\begin{array}{c} {\langle F\rangle }_{M,N}=\sum_{n}P\left(n\right)F\left(n\right)=\frac{1}{M-N}\sum_{n}P\left(n\right)\sum_{i=2}^{\infty }\sum_{j=1}^{i-1}n_{i}F_{i-j,j}, \end{array}$$
where P(n) is the probability of distribution and the outer summation is over all distributions in ensemble $E_{M,N}$.

### 3.2. Fragmentation Ensemble

To construct the ensemble of distributions in fragmentation we begin with a single particle with mass *M* and subject it to *g* fragmentation events, an event consisting of replacing a cluster with two fragments. After *g* events the distribution contains N=g+1 elements (clusters). The set of all distributions that can be reached in *g* fragmentation events forms the fragmentation ensemble $E_{M,N}$. All distribution in the ensemble satisfies the two constraints in Equation (Equation 1) and, conversely, all distributions that satisfy these constraints can be formed by binary breakage of mass *M* in g=N−1 fragmentation steps.

We represent this process pictorially in the form of a layered graph. Figure 1 illustrates this graph for M=7. Nodes represent fragment distributions and edges indicate transitions between distributions following a fragmentation event. The graph is layered the number of fragmentation events to reach a distribution and each layer contains the complete ensemble of distributions that can be reached in fixed number of fragmentation events. The graph is the exact reverse of that of aggregation: by reversing the direction of all arrows we obtain the graph for discrete binary aggregation of *M* monomers (bottom) advancing towards a single cluster (top) through M−1 binary aggregation events [19].

### 3.3. Transitions between Distributions

When a cluster with mass *i* in distribution $n^{\prime }$ of ensemble $E_{M,N-1}$ breaks up into fragments i−j and *j* the result is a distribution n in the ensemble of the next generation, $E_{M,N}$; schematically,
(14)$$\begin{array}{c} n^{\prime }\overset{\left(i)\to (i-j)+(j\right)}{\to }n. \end{array}$$ The probability of this transition is proportional to the number $n_{i}^{\prime }$ of clusters with mass *i* in the parent distribution and to the fragmentation rate $F_{i-j,j}$:(15)$$\begin{array}{c} P\left(n^{\prime }\to n\right)=\frac{n_{i}^{\prime }F_{i-j,j}}{\left(M-N+1\right){\langle F\rangle }_{M,N-1}}. \end{array}$$ Here, ${\langle F\rangle }_{M,N-1}$ is the mean fragmentation kernel over all fragmentation events in the parent ensemble:(16)$$\begin{array}{c} {\langle F\rangle }_{M,N-1}=\frac{\sum_{n^{\prime }}P\left(n^{\prime }\right)\sum_{i=2}^{\infty }\sum_{j=1}^{i-1}n_{i}^{\prime }F_{i-j,j}}{M-N+1}, \end{array}$$
where $P(n^{\prime })$ is the probability of distribution in the parent ensemble. There are M−N+1 ordered pairs of fragments in any distribution that contains N−1 clusters with total mass *M*. Accordingly, $\left(M-N+1\right){\langle F\rangle }_{M,N-1}$, which appears as the normalizing constant in the transition probability in Equation (Equation 15), is the total fragmentation rate in parent ensemble $E_{M,N-1}$.

The probability of distribution propagates from one generation to the next with transition probability $P(n^{\prime }\to n)$ according to the Master Equation,
(17)$$\begin{array}{c} P\left(n\right)=\sum_{n\prime }P\left(n^{\prime }\to n\right)P\left(n^{\prime }\right), \end{array}$$
with the summation taken over all parents $n^{\prime }$ of distribution n; in terms of Figure 1 the summation is over all edges of the fragmentation graph that lead to n. At the initial state distribution $n_{0}$ consists of a single cluster with mass *M*. In this state, we have
(18)$$\begin{array}{c} n_{0}!=W\left(n_{0}\right)=P\left(n_{0}!\right)=\Omega_{M,1}=1. \end{array}$$ With Equation (Equation 18) as the initial condition, the Master Equation fixes the probabilities of all distributions in all future generations. Expressing the probability of parent and offspring distributions in terms of the sampling functional and the partition function, and using Equation (Equation 15) for the transition probabilities the Master Equation produces the following result
(19)$$\begin{array}{c} \frac{\Omega_{M,N-1}}{\Omega_{M,N}}=\frac{N-1}{\left(M-N+1\right){\langle F\rangle }_{M,N-1}}\left\{\sum_{i=2}^{\infty }\sum_{j=1}^{i-1}\frac{n_{i-j}\left(n_{j}-\delta_{i-j,j}\right)F_{i-j,j}}{N(N-1)}\frac{W(n^{\prime })}{W(n)}\right\}. \end{array}$$ To obtain this result, we used the fact that the parent-offspring relationship is reversible (Figure 1): if parent distribution $n^{\prime }$ produces offspring n via the fragmentation event (i)→(i−j)+(j), this parent can be constructed by applying the aggregation reaction (i−j)+(j)→(i) to the offspring distribution. In view of this observation, each element of the double summation in Equation (Equation 19) is a fragmentation parent of n that contributes towards the offspring distribution with rate $F_{i-j,j}$. Accordingly, the double summation goes over all parents of offspring distribution n. Since the left-hand side is independent of distribution n, so must be the right-hand side. This can be true only if the quantity in braces is independent of distribution n,
(20)$$\begin{array}{c} \sum_{i=2}^{\infty }\sum_{j=1}^{i-1}\frac{n_{i-j}\left(n_{j}-\delta_{i-j,j}\right)F_{i-j,j}}{N(N-1)}\frac{W(n^{\prime })}{W(n)}=\Lambda . \end{array}$$ We fix Λ by imposing a normalization condition on the sampling functional: if W(n) is constant for all distributions, we require W(n)=1. We will confirm in Section 4 that if $F_{i,j}=1$ then W(n)=const. This implies Λ=1. Accordingly Equation (Equation 19) separates into two independent recursions, one for the partition function and one for the sampling functional. For the partition function, we have
(21)$$\begin{array}{c} \frac{\Omega_{M,N-1}}{\Omega_{M,N}}=\frac{N-1}{\left(M-N+1\right){\langle F\rangle }_{M,N-1}}, \end{array}$$
which is readily solved for $\Omega_{M,N}$ starting with $\Omega_{M,1}=1$ in generation 0. The result is
(22)ΩM,N=M−1N−1∏N′=1N−1FM,N′. The corresponding expression for the sampling functional is
(23)$$\begin{array}{c} W\left(n\right)=\sum_{i=2}^{\infty }\sum_{j=1}^{i-1}\frac{n_{i-j}\left(n_{j}-\delta_{i-j,j}\right)F_{i-j,j}}{N(N-1)}W\left(n^{\prime }\right), \end{array}$$
which gives the sampling functional of distribution in terms of the sampling functional of its parents. The parameters β and *q* are
(24)$$\begin{array}{c} \beta =\log\frac{\Omega_{M+1,N}}{\Omega_{M,N}}=\log\left(\frac{M}{M-N+1}\right)+\sum_{N^{\prime }=1}^{N-1}\log\frac{{\langle F\rangle }_{M+1,N^{\prime }}}{{\langle F\rangle }_{M,N^{\prime }}}, \end{array}$$
(25)$$\begin{array}{c} q=\frac{\Omega_{M,N+1}}{\Omega_{M,N}}=\frac{M-N}{N}{\langle F\rangle }_{M,N} \end{array}$$
and are obtained by applying the finite-difference form of the derivatives in Equation (Equation 6) to the partition function in Equation (Equation 22).

### 3.4. Kinetics and Thermodynamics

We may now summarize the key results of the stochastic theory of fragmentation. Equations (Equation 22) and (Equation 23) represent *kinetic* properties of the ensemble and embody the transition probabilities in all possible trajectories in the event space of distributions. One additional kinetic result is obtained in Section 3.5 where we derive the classical fragmentation equation. Equations (Equation 22)–(Equation 25) represent *thermodynamic* properties. The partition function characterizes the ensemble of feasible distributions; the parameters β and *q* are the derivatives of the partition function and appear in the most probable distribution in Equation (Equation 5). The factors $w_{k}^{*}$, which are also needed in in Equation (Equation 5), are to be obtained from the derivatives of the sampling functional according to Equation (Equation 6). In the context of this theory, the mathematical problem reduces to the calculation of Ω and logW. We demonstrate this procedure in Section 4 with an example that produces analytic results for the partition function and the sampling functional.

### 3.5. Mean Distribution

The mean number of clusters with mass *k* is
(26)$$\begin{array}{c} \langle n_{k}\rangle =\sum_{n}n_{k}P\left(n\right), \end{array}$$
where $n_{k}$ is the *k*th element of distribution n, P(n) is the probability of distribution, and the summation is over all n in the ensemble. Using P(n) from Equation (Equation 17) and the stoichiometry of the fragmentation reaction we obtain
(27)$$\begin{array}{c} {\langle n_{k}\rangle }_{M,N}=\sum_{n}\sum_{i=2}^{\infty }\sum_{j=1}^{i-1}P\left(n^{\prime }\to n\right)\left(n_{k}^{\prime }-\delta_{k,i}+\delta_{k,i-j}+\delta_{k,j}\right)P\left(n^{\prime }\right)=\\ {\langle \sum_{i=2}^{\infty }\sum_{j=1}^{i-1}P\left(n^{\prime }\to n\right)\left(n_{k}^{\prime }-\delta_{k,i}+\delta_{k,i-j}+\delta_{k,j}\right)\rangle }_{M,N-1}. \end{array}$$ Here primed terms refer to parent elements of element $n_{k}$ and the ensemble average on the right-hand side is over the parent ensemble $E_{N,N-1}$. Lastly, we use Equation (Equation 15) for the transition probability P and perform the summations over the Kronecker deltas. The final result is
(28)$$\begin{array}{c} {\langle n_{k}\rangle }_{M,N}-{\langle n_{k}^{\prime }\rangle }_{M,N-1}=\frac{{\langle -\sum_{j=1}^{k-1}n_{k}^{\prime }\;F_{k-j,j}+2\sum_{j=k+1}^{\infty }n_{j}^{\prime }\;F_{j-k,k}\rangle }_{M,N-1}}{\left(M-N+1\right){\langle F\rangle }_{M,N-1}}. \end{array}$$ This is the governing equation for the evolution of the mean distribution: on the left-hand side, we have the change in the mean number of cluster of size *k* between generations; on the right-hand side, we have the average over all trajectories emanating from the the parent generation.

If the ensemble is represented by a single distribution, the *mean* distribution $\bar{n}$, the result can be written as as difference equation for ${\bar{n}}_{k}$ with the ensemble averages dropped:(29)$$\begin{array}{c} \frac{\Delta {\bar{n}}_{k}}{\Delta N}=\frac{-\sum_{j=1}^{k-1}{\bar{n}}_{k}\;F_{k-j,j}+2\sum_{j=1k+1}^{\infty }{\bar{n}}_{j}\;F_{j-k,k}}{\left(M-N+1\right)\bar{F}}. \end{array}$$ Here, ΔN=1 is the change in the number of clusters between successive generations and $\bar{F}$ is the mean kernel over all fragmentation events in $\bar{n}$. To make full connection with the classical treatment we write the rate equation for the number of clusters *N*. Each binary fragmentation event produces one net new cluster, therefore, the rate at which the parent generation produces clusters is
(30)$$\begin{array}{c} \frac{\Delta N}{\Delta t}=\sum_{n\prime }P\left(n^{\prime }\right)\sum_{i=2}^{\infty }\sum_{j=1}^{i-1}n_{i}^{\prime }F_{i-j,j}=\left(M-N+1\right){\langle F\rangle }_{M,N+1}. \end{array}$$ Dividing Equation (Equation 29) by (Equation 30) we obtain
(31)$$\begin{array}{c} \frac{\Delta {\bar{n}}_{k}}{\Delta t}=-\sum_{j=1}^{k-1}{\bar{n}}_{k}\;F_{k-j,j}+2\sum_{j=k+1}^{\infty }{\bar{n}}_{j}\;F_{j-k,k}. \end{array}$$ We recognize the result as the classical fragmentation equation for the mean distribution. It is the appropriate limit of of Equation (Equation 28) if the probability of distribution, P(n), is sharply peaked around a single distribution. In this case, and in this only, all average quantities over the ensemble can be replaced by the same quantity evaluated over a single distribution, the most probable distribution of its generation, also equal to the mean distribution in the ensemble.

## 4. Special Case: Random Fragmentation $\left(F_{i-j,j}=1\right)$

The constant kernel, $F_{i-j,j}=1$, represents random fragmentation, a process in which all possible fragmentation events are equally probable. The fragmentation rate is $a_{k}=k-1$, equal to the number of possible events within mass *k*, and the fragment distribution is uniform, b(i|k)=2/(k−1). For this model we obtain the full solution of the stochastic problem in closed form.

The mean kernel in any distribution n is F(n)=1 and it follows that ${\langle F\rangle }_{M,N}=1$ for all *M*, *N*. The partition function follows from Equation (Equation 22)
(32)ΩM,N=M−1N−1,
and is equal to the number of ways to break mass *M* into *N* ordered fragments [21]. To obtain the sampling functional we note the identity
(33)$$\begin{array}{c} \sum_{i=2}^{\infty }\sum_{j=1}^{i-1}\frac{n_{i-j}\left(n_{j}-\delta_{i-j,j}\right)}{N(N-1)}=1. \end{array}$$ We conclude from Equation (Equation 23) that W(n) must be the same for all distributions in all generations, and since $W(n_{0})$ in generation zero, we obtain
(34)$$\begin{array}{c} W(n)=1. \end{array}$$
with $w_{i}=1$. This is a special case of functionals that factorize in the form
(35)$$\begin{array}{c} W\left(n\right)=\prod_{i=1}^{\infty }w_{i}^{n_{i}}. \end{array}$$ Such systems, also known as Gibbs ensembles, appear in various probabilistic contexts and their properties are well known [22,23,24]. In particular, the mean cluster distribution is given in terms of the partition function and the factors $w_{k}$ by [17]
(36)nkM,NN=wkΩM−k,N−1ΩM,N=M−k−1N−2/M−1N−1. The derivatives β and *q* are
(37)$$\begin{array}{c} \beta =\log\left(\frac{M}{M-N+1}\right)∼\log\left(\frac{M}{M-N}\right) \end{array}$$
(38)$$\begin{array}{c} q=\frac{M-N}{N} \end{array}$$
and the most probable distribution is exponential:(39)$$\begin{array}{c} \frac{n_{k}^{*}}{N}=\frac{e^{-\beta k}}{q}=\frac{1}{\bar{x}-1}{\left(1-\frac{1}{\bar{x}}\right)}^{k} \end{array}$$
where $\bar{x}=M/N$ is the mean cluster size. It is easy to show that the mean distribution in Equation (Equation 36) and the most probable distribution converge to each other.

Equation (Equation 36) for the mean fragment distribution is exact for all integer masses M>N and fragment masses M−N+1≥k≥1. The first few fragment distributions for general *M* are shown in Table 1. For N=2 the distribution is uniform; for N=3 it is linear in fragment size; in general the distribution is a polynomial of degree N−1 and always monotonically decaying in the size range (0,M). It is interesting to point out that these distributions are the same as those produced by a single random fragmentation event into *N* pieces (see Equation (Equation 12) in [21]). That is, *g* consecutive random binary fragmentation events produce the same ensemble of fragments as as a single random fragmentation event into g−1 pieces. These distributions are also the same as those in discrete binary aggregation with constant aggregation kernel $K_{i,j}=1$ [19]. That is, fragmentation with $F_{i,j}=1$ and aggregation with $K_{i,j}=1$ both assign the same probabilities to all distributions int he event space, even though the two processes move in opposite directions with different transition probabilities each. This symmetry between fragmentation and aggregation has not been recognized previously.

As a demonstration of the exactness of Equation (Equation 36) we perform a Monte Carlo simulation of random fragmentation with M=50 (Figure 2). The agreement between simulation and the theoretical distribution is excellent. (Details on the Monte Carlo simulation are given in Section 5.3).

## 5. Shattering

### 5.1. Power-Law Breakage

A special family of fragmentation kernels is of the power-law form
(40)$$\begin{array}{c} F_{i,j}={\left(i+j-1\right)}^{\nu }, \end{array}$$
where ν is a constant. This kernel represents binary breakage with power-law rate $a_{k}={\left(k-1\right)}^{\nu +1}$ and uniform fragment distribution b(i|k)=2/(k−1). The power-law kernel has been studied extensively, especially since it leads to shattering when the exponent is negative [5,6,7,8,16]. From a kinetic perspective, shattering is the result of the accelerated fragmentation rate with decreasing size that produces an accumulation at the low end of the distribution. Here, we examine shattering under the criteria for thermodynamic stability. In molecular thermodynamics phase splitting occurs when the concavity of the microcanonical partition function is violated. The same condition identifies the presence of shattering, as we will see.

The mean fragmentation rate of the power-law kernel in distribution n is
(41)$$\begin{array}{c} F\left(n\right)=\frac{1}{M-N}\sum_{i=2}^{\infty }\sum_{j=1}^{i-1}n_{i}{\left(i-1\right)}^{\nu }=\frac{1}{M-N}\sum_{i=2}^{\infty }n_{i}{\left(i-1\right)}^{\nu +1}∼\frac{N}{M-N}{\left(\frac{M}{N}\right)}^{\nu +1}∼{\left(\frac{M}{N}\right)}^{\nu }, \end{array}$$
when M≫N≫1. Thus we obtain the scaling
(42)$$\begin{array}{c} F\left(n\right)∼{\langle F\rangle }_{M,N}∼{\left(\frac{M}{N}\right)}^{\nu }. \end{array}$$ We may now obtain the partition function. Using Equation (Equation 22) we find
(43)ΩM,N=M−1N−1MN−1(N−1)!ν−1. Its parameters β and *q* are obtained from the derivatives of the partition function according to Equation (Equation 6). The result can be expressed as
(44)$$\begin{array}{c} \beta =-\log\theta +(\nu -1)(1-\theta ), \end{array}$$
(45)$$\begin{array}{c} q=\theta {\left(1-\theta \right)}^{-\nu }, \end{array}$$
where θ is a scaled size variable, defined as
(46)$$\begin{array}{c} \theta =1-\frac{N}{M}. \end{array}$$ The scaled variable maps the complete evolution of the system into the interval 0≤θ≤1: Fragmentation commences at θ=1−1/M∼1 and ends at θ=0.

### 5.2. Stability and Phase Transitions

Stability requires $\log\Omega_{M,N}$ to be concave function of *M* and *N*. The condition ensures that the most probable distribution is the solution to $\partial \log P\left(n^{*}\right)/\partial n^{*}=0$ and is given by Equation (Equation 5). Concavity requires the second derivatives of $\log\Omega_{M,N}$ to be negative. In terms of the scaled variable θ the conditions reduce to the concave conditions are
$$\begin{array}{c} {\left(\frac{\partial^{2}\log\Omega }{\partial M^{2}}\right)}_{N}\leq 0\Rightarrow {\left(\frac{d\beta }{dM}\right)}_{N}\leq 0\;\mathrm{and}\;{\left(\frac{\partial^{2}\log\Omega }{\partial N^{2}}\right)}_{M}\leq 0\Rightarrow {\left(\frac{d\log q}{dN}\right)}_{M}\leq 0. \end{array}$$ Expressing *M* or *N* in terms of θ=1−N/M we obtain
(47)$$\begin{array}{c} \frac{d\beta }{d\theta }\leq 0;\;\frac{d\log q}{d\theta }\geq 0. \end{array}$$ Using Equations (Equation 44) and (Equation 45), both inequalities in Equation (Equation 47) reduce to the same result:(48)$$\begin{array}{c} \theta \leq \frac{1}{1-\nu }. \end{array}$$ This condition guarantees concavity of the partition function. In the language of thermodynamics, it guarantees the stability of a single-phase system; if it is violated the system splits into two distinct coexisting phases. The stability maps of power-law fragmentation are shown in Figure 3 on the (β,θ) and (q,θ) planes. For ν>0 the system is always stable and is represented by a single phase, a distribution of clusters given by Equation (Equation 5). When ν is negative stability depends the critical value $\theta^{*}=1/\left(1-\nu \right)$ falls within the interval (0,1) and splits it into two regions. Below $\theta^{*}$ the system is stable, above $\theta^{*}$ is not. As fragmentation always starts at θ=1, the system begins in the unstable regime: shattering is observed at all times.

### 5.3. Monte Carlo Simulations of Shattering

The most direct way to illustrate the shattering instability is through simulation. We use Monte Carlo to sample trajectories in the phase space of fragmentation. The simulations are conducted as follows. Given *N* clusters we choose a cluster to break with probability proportional to ${\left(k-1\right)}^{\nu }$, where *k* is the mass of the cluster. Mass *k* is then split into integer fragments *i* and k−i, which are chosen with equal probability among all possible k−1 fragments. The simulation begins with a single cluster with mass *M*, which is then subjected to M−1 fragmentation events until it forms *M* fragments with unit mass. This constitutes one trajectory that tracks the distribution of fragments from generation g=0 to generation g=M−1. The mean distribution is calculated by averaging 5000 trajectories.

Figure 4 shows results for three different masses, M=50, 100 and 200, to highlight the convergence to asymptotic behavior. Cluster distributions are plotted against the scaled mass z=i/M. In the limit M→∞ this scaling amounts to breaking a particle of unit mass into fragments in the continuous interval from 0 and 1. For ν=1 (random fragmentation), the system is stable at all times and its distribution is given by Equation (Equation 39). The results of the simulation in Figure 4a–c For ν=−3, the system is unstable and the distribution of clusters contains a new feature: the number of the smallest fragment size (unit mass) deviates markedly above the distribution of the rest of the fragments. This feature persists: it does not go away as *M* is increased, neither does is subside as fragmentation goes on. In fact, the divergence of this segment of the population becomes more pronounced as M→∞. In this limit the divergent mass is $z^{*}=1/M\to 0$: accordingly, a finite fraction of the mass of the system is accumulated into fragments of zero size.

## 6. Discussion

The emergence of a ghost fraction of particles (“dust”) with zero mass has been termed “shattering” by McGrady and Ziff [6] and in the asymptotic limit manifests itself as a breakdown of mass conservation. It is characteristic of kernels that preferentially favor the fragmentation of smaller sizes, producing a current of mass to the monomers. The power-law kernel with negative exponent is of this type. This kernel was studied by Ziff and coworkers [6,16] who discussed shattering as a phase transition analogous to gelation in aggregation. Our treatment provides a rigorous basis to these analogies, which up to now were were drawn qualitatively: shattering *is* a phase transition. Ir occurs when the partition function violates the criterion of thermodynamic stability.

McGrady and Ziff [6] pointed out the close connection between shattering and gelation. Aggregation with power-law kernel $K_{i,j}={\left(ij\right)}^{\alpha /2}$ is analogous to the power-law fragmentation kernel $F_{i,j}={\left(i+j\right)}^{\nu }$: it produces stable solutions for α≤1 but leads to gelation for α>1. This connection is not merely qualitative. The parameters β and *q* for power-law aggregation were obtained by [19]. We compare them to those for power-law fragmentation (Table 2). With the substitution α=1−ν, the parameter *q* and the region of stability are the same in both processes. Gelation occurs when a cluster appears in the giant region (M−N+1)/2+1<kM−N+1, a region defined by the condition that it may contain at most one cluster [4]. In the scaling limit of aggregation (M>N→∞ at fixed M/N) the giant cluster goes over to infinity and disappears from the size distribution, giving rise to an apparent loss of mass and the breakdown of mass conservation. The giant cluster is also present in shattering. In terms of the scaled mass z=i/M, the giant cluster in fragmentation is asymptotically of the order 1. Under shattering conditions (Figure 4d–f) the neighborhood of z≈1 remains populated at all times. By contrast, stable solutions are devoid of clusters in this region (Figure 4a–c). The presence of a giant cluster is a universal feature of instability, whether the process is aggregation or fragmentation.

Fundamentally shattering and gelation represent the same condition: the presence of a single cluster in the giant region that coexists with a distribution of clusters in the sub-giant region. It is the different scaling between fragmentation and aggregation that sends the ghost phase to zero size (shattering) or infinite size (gelation), where it becomes invisible either in the form of an infinite concentration of clusters with zero mass (shattering) or as a zero concentration of particles with infinite mass. There is, however, one difference: in gelation the system begins in the stable branch and within finite time crosses into the unstable regime. As we have shown previously [4,19] in this case we can construct the tie line between the two phases by standard thermodynamic arguments to obtain the distribution of the dispersed phase and the gel fraction at all times. In shattering, the system begins in the unstable region and never crosses over to the stable regime. Though we believe that in this case also it ought to be possible to construct the tie line, we were not able to do it here. This requires a more detailed treatment of the unstable partition function, in particular, the asymptotic scaling of the mean fragmentation kernel in Equation (Equation 42) when the exponent is negative. In discrete finite systems instability cannot occur instantaneously. The initial state is stable and it must take a finite number of fragmentation events to reach instability. The scaling form of the mean kernel in Equation (Equation 42), while sufficient to map the stability boundary, it is not of sufficient accuracy to study the onset of phase splitting.

## Figures and Tables

**Figure 1 entropy-24-00229-f001:**
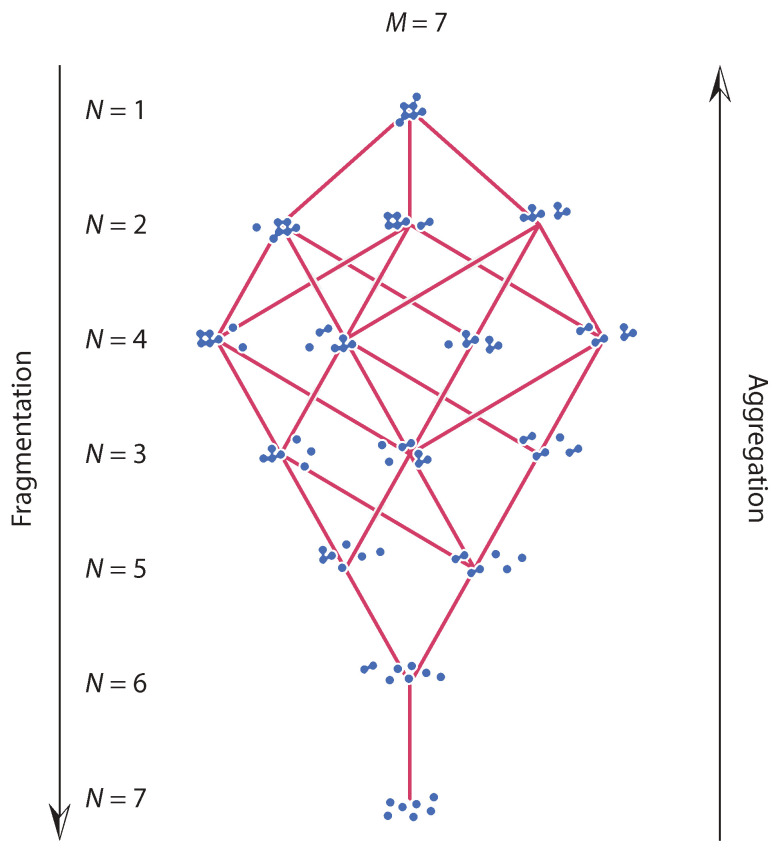
The graph of discrete binary breakage of mass M=7. By reversing the direction of transitions we obtain the graph of discrete binary aggregation starting with M=7 monomers. A continuous path from top to bottom represents one possible trajectory through the space of feasible distributions.

**Figure 2 entropy-24-00229-f002:**
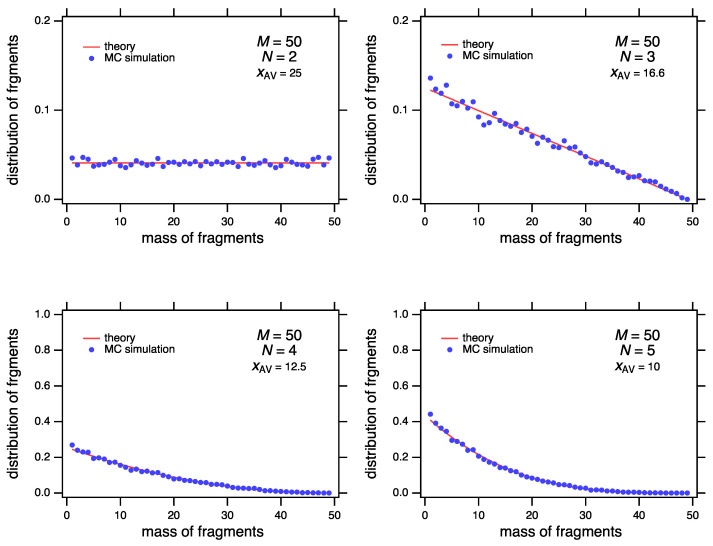
Random fragmentation of M=50 into N=2,3,4 and 5 pieces. The results from the Monte Carlo simulation are in full agreement with the theoretical distribution of fragments, Equation (Equation 36), also in Table 1.

**Figure 3 entropy-24-00229-f003:**
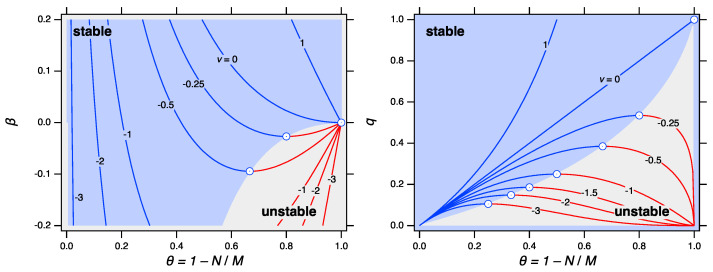
Stability map of power-law kernel in terms of the parameters β (**left**) and *q* (**right**). Stable branches are shown blue, unstable in red.

**Figure 4 entropy-24-00229-f004:**
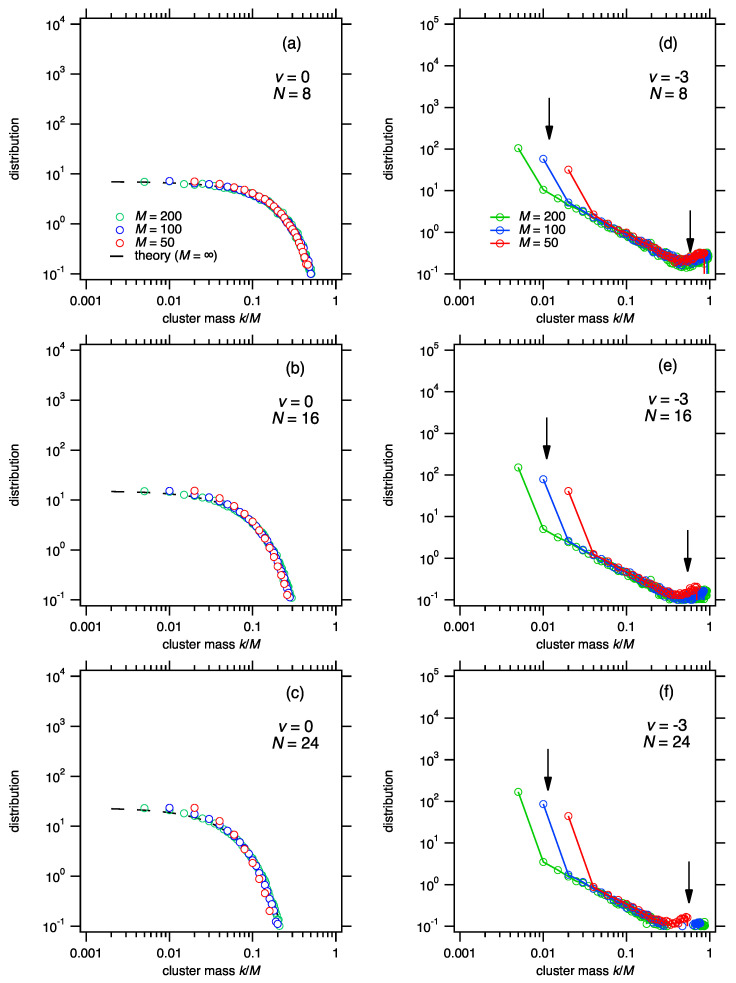
Cluster size distributions by Monte Carlo simulations of breakup with $F_{i,j}={\left(i+j\right)}^{\nu }$. (**a**–**c**) ν=0: The system is stable and its distribution is given by Equation (Equation 36). The concentration of particles with scaled mass z=k/M∼1 is zero. (**d**–**f**) ν=−3: The system is unstable and undergoes shattering. Distributions in the scaled cluster size z=k/M are characterized by the simultaneous presence of cluster masses of the order 0 and of the order 1 (arrows).

**Table 1 entropy-24-00229-t001:** Mean distributions of random binary fragmentation with constant kernel $F_{i,j}=1$ for N=2,3,4 and 5.

Number of Fragments, *N*	Distribution of Fragments
2	$\frac{1}{M-1}$
3	$\frac{2(M-k-1)}{\left(M-2)(M-1\right)}$
4	$\frac{3(M-k-2)(M-k-1)}{\left(M-3)(M-2)(M-1\right)}$
5	$\frac{4(M-k-3)(M-k+-2)(M-k-1)}{\left(M-4)(M-3)(M-2)(M-1\right)}$

**Table 2 entropy-24-00229-t002:** Comparison between power-law fragmentation and power-law aggregation.

	Fragmentation	Aggregation
kernel	${\left(i+j\right)}^{\nu }$	${\left(ij\right)}^{\alpha /2}$
β	−logθ+(ν−1)(1−θ)	−logθ+αθ
*q*	$\theta {\left(1-\theta \right)}^{-\nu }$	$\theta {\left(1-\theta \right)}^{\alpha -1}$
stability	ν≥0	α≤1

## Data Availability

No data used in this study.

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
