# Peer review of "Stochastic Theory of Discrete Binary Fragmentation—Kinetics and Thermodynamics"

_entropy, 2022, doi:10.3390/e24020229_

Round 1

Reviewer 1 Report

This manuscript revisits the problem of binary fragmentation and investigates what is termed the "Statistical Ensemble" of binary fragmentation.  As someone who has worked on fragmentation problems in the past, I had a hard time seeing what was really new in the manuscript.  The models treated by the author have all been solved in previous articles (as cited by the author), and I didn't see what new insights were gained by the present manuscript.

One new result seems to be Eq. (26), which gives the fragment size distribution after a fixed number of breakup events.  This seems to be a simple combinatoric exercise that looks more complicated than it should be because the result is derived via the complicated formalism of this manuscript.  By viewing the fragmentation process as repeated cutting of the line interval, I am sure the result (26) can be derived straightforwardly and with not much technical complications.  I don't see the value of the complicated approach presented here.

Another possibly new result is contained in the stability diagrams of Fig. 3.  I have to admit not knowing if this was truly new or whether is was a complicated way of presenting already known results.  I'm also unsure about the stability criterion itself.  In the shattering regime, why physically does does the most probable distribution satisfy the condition d[ln P(n)]/dn*=0?

Let me also provide some detailed comments:

The authors states that an integer mass splits into two fragments of integer mass.  This is not appropriate if the author is going to be treating the shattering transition.  My impression is that the author is viewing a cluster as consisting of unit-mass monomers and that a breakup event involves removing links between some constituent monomers, so that the two fragments are smaller clusters that again consist of monomer.  Please clarify the nature of the model.

What is the point of the mean fragmentation kernel F(n)?  I don't see that it provides useful insight or a way to solve any fragmentation model.

In Eq. (16), it's unfortunate to use the letter P to denote two different quantities. Please fix the notation.

Figure 1 is a bit confusing and it took me a while to realize that the different separations between the various dots was deliberate and tried to indicate distinct clusters.  Either the separation between clusters should be larger or perhaps use different colors for different clusters.

It seems that Fig. 3 is not referred to in the text. I guess there is a missing "3" in the line after line 207

There is a mismatch between Fig. 4 and the caption.  The caption mentions nu=-1, but the right figure panels indicate nu=-2.  Which is right?

Finally, there are lots of minor and sometimes annoying errors and typos.  There are too many to enumerate and the author should do a careful reading of the manuscript to correct them.

To summarize, this manuscript gives a rather formal way of treating the classic fragmentation problem through the lens of statistical thermodynamics.  I am not sure what one physically learns that is truly new by this approach.  I admit to not fully appreciating all the results given in the manuscript, but I don't have the patience to learn the statistical thermodynamic approach because it seems that it doesn't provide new insights beyond what learns by solving the rate equations for fragmentation.

Reviewer 2 Report

Entropy-1534689

The Statistical Ensemble of Binary Fragmentation, by Themis Matsoukas

The author presents a Statistical model for fragmentation, based on previous work also published in Entropy: 10.3390/e21090890, 10.3390/e22101181. The manuscript has valuable information, particularly in the way in which gelation is described. It would be desirable that the author could explain in more detail the novelity of the approach discussed in this paper in relation to the references already given of his previous work in Entropy. On the other hand, the description of the Monte Carlo simulations is not complete (method, selection of simulation parameters, number of cycles, size effects, uncertainties, etc.). 

I have found misprints in lines 32, 33, 138, 139, 148, 187.

Equation (4) is not correct (see section 3.2, before equation (14) in  10.3390/e21090890). 

Equation (29): x is not defined.

Reviewer 3 Report

This paper studies binary fragmentation processes from a thermodynamic point of view, in the sense that to any fragment size distribution is associated a probability, with the constraint of constant mass M and number of fragment N. The shattering transition previously seen in kinetic fragmentation models is interpreted here as a breakdown of stability, or a change in the concavity of the associated partition function. Some comparisons are drawn with the gelation transition in binary aggregation processes. Interestingly, in fragmentation with shattering, the process starts right in the unstable branch, whereas it starts in the stable one in aggregation, before reaching the unstable branch at a finite time. This is an interesting paper, which reveals new insights on fragmentation processes. 

As a major concern, I find the paper very hard to read because of being imprecise on several important points, and for containing too many typos. The author should really check all his equations carefully again.

1) Define explicitly \Omega_{M,N}, as this function will be used later. Is it the number of different ways of obtaining N fragments with total mass M? 

2) There is a problem with (11): the r.h.s. of (11) does not depend on n, which is strange. I guess that there is a n_i missing, which is present in (12). In addition, in the upper limit of the first sum: infty->M? The lower limit should be 2, not 1, to be consistent with (12).

What is the definition of a fragmentation event? The lines after (11) are not clear. The number of possible broken bound in a fragment of size n_i is n_i-1. Hence the normalization constant over all fragments is sum_i^N (n_i-1)=M-N

3) Relative to the discussion of Fig. 1, it should be recalled that kinetically, the fragmentation and aggregation processes are not truly  reverse one from each other: fragmentation is a linear process, whereas aggregation in nonlinear.

4) Eq. (15) is a repetition of (12) in another notation, which is confusing. Eq. (15) seems to be (12), changing N by N-1. If so, rather define <F>_{M,N} right away in (12).

5) Give more details on how (17) was obtained. In addition, shouldn’t it be n’_{i-j} and n’_j instead of n_{i-j} and n_j in the sum? Same comment for (20). If (20) is true as written, then W(n’) should be factorized out of the sum.

6) Between (17) and (18): “For the same reasons as discussed in Matsoukas [18] for the case of aggregation…” Explain these reasons here again.

7) Give the intermediate step(s) allowing to obtain the unnumbered equation between (23) and (24).

8) Eq. (26): The average <n_k> has not been defined. It would be better to first derive n^*_k and then <n_k> (after defining it). 

Actually, the author should detail how w^*_k in (6) was calculated in this case in order to obtain the most probable distribution (29). From its definition, it seems that w^*_k=0, not 1. The notation w_i in (25) is also very confusing: if one applies (6) to (25), it gives w^*_i=ln (w_i), which is 0 if w_i=1.

9) After (30): a_k=(k-1)^{\nu+1} instead of k^{\nu+1}?

10) In Eq. (31): n_i*i^{\nu} should be n_i*(i-1)^{\nu}, I presume
The sum should also start at i=2 for consistency. This part is totally confusing regarding the kernels.

11) “Since F(n) is the same in every distribution..”. Rather, (32) holds true only for the typical distribution, not for F(n) with any n.

12) How the ~ sign in Eq. (32)-(33) became a = sign in (34)?

13) Which equations are used to obtain (37)-(39)?

14) Footnote 1: “…the derivative of b changes sign whereas the sign of the derivative of log q remains unchanged.” It is not the other way around?

Other typos: 

-Introduction : “In this paper we apply this formulation to irreversible binary aggregation” -> fragmentation?

-Eq. (4) should read: log[W(lamnda n)]=lambda ln[W(n)], I presume.

-After Eq. (8): nay->may.

-Between (11) and (12): assemble—>ensemble.

-Last sentence before 3.3: ensemble—>distributions?

-Eq. (14): P(n’—>n).

-Before (18): Fro—>For.

-After (22): 19—>(19). Eq. (22) is not part of the “thermodynamic treatment”?

-After (29): “for N = 3 it is linear…”

-Same paragraph: “those produces”—>those produced.

-After (4): Figure number missing in text.

-“What are these phases? This question is best answered by first inspecting the results of simulation.” simulations

-Discussion, first paragraph: “is z”
